# Deciphering oxidative stress contributions in vestibular schwannoma: A bioinformatics approach to novel therapeutic pathways

Yubin Xue[ID][1][☯]*, Mingyue Wang[2][☯], Hongwei Ma[3]

1 Department of Otorhinolaryngology, Beijing Tiantan Hospital, Capital Medical University, Beijing, China, 2 Department of Publicity, The First People's Hospital of Shizuishan, Ningxia Medical University, Shizuishan, Ningxia, China, 3 Department of Science and Education, The First People's Hospital of Shizuishan, Ningxia Medical University, Shizuishan, Ningxia, China

☯ These authors contributed equally to this work.
* xueyubin2000@163.com

## Abstract

### Background

Vestibular schwannoma (VS) is a benign tumor originating from Schwann cells, and its molecular pathogenesis remains poorly understood. Increasing evidence suggests oxidative stress (OS) plays a critical role in tumor development, but its involvement in VS is largely unexplored.

### Methods

We analyzed two GEO transcriptomic datasets (GSE54934 and GSE56597) to identify oxidative stress-related differentially expressed genes (OSRDEGs). Functional enrichment, protein–protein interaction (PPI) network construction, hub gene identification, and immune infiltration analyses were performed to uncover potential molecular mechanisms.

### Results

Fifteen OSRDEGs were identified, and nine hub genes (IL6, CYBB, CAV1, EGFR, SELE, IL18, CDKN2A, ADIPOQ, CDH2) were screened. Enrichment analysis indicated that these genes are mainly involved in apoptosis, reactive oxygen species regulation, and immune-related pathways. Moreover, immune infiltration analysis revealed significant differences in CD8＋T cells and macrophage populations between VS and control tissues.

### Conclusions

Our study suggests that oxidative stress may contribute to VS progression by influencing immune responses and signaling pathways. These findings provide new

**Data availability statement:** Data Availability Statement The minimal dataset underlying this study is openly available at Zenodo (DOI: https://doi.org/10.5281/zenodo.17163233). Code Availability All author-generated analysis code used in this study is archived together with the dataset at Zenodo (DOI: https://doi.org/10.5281/zenodo.17163233).

**Funding:** This study was supported by the Health and Family Planning Commission of Ningxia Hui Autonomous Region through the Health Scientific Research Project – Youth Training Program (Grant No. 2025-NWQP-B076, awarded to Yubin Xue). The funder's website is available at http://wsjkw.nx.gov.cn/.

**Competing interests:** The authors have declared that no competing interests exist.

insights into the molecular basis of VS and may guide future experimental and therapeutic investigations.

---

# 1 Introduction

Vestibular schwannoma (VS), predominantly originating from schwann cells of the vestibular nerve's sheath within the internal auditory canal, stands as the most frequent tumor in the pontocerebellar angle region, constituting approximately 8% of all intracranial tumors. The lifetime risk of developing this condition exceeds 1 in 500, and incidence rates annually shows a range of 2.2 to 5.2 per 100,000 individuals [1–3]. In its initial stages, VS predominantly manifests through symptoms associated with the impairment of the eighth cranial nerve, including hearing loss, tinnitus, and balance disorders. As the tumor expands, exerting pressure on the brainstem and adjacent nerves, it may induce facial paralysis, trigeminal nerve innervation area hypesthesia, trigeminal neuralgia, headaches, and hydrocephalus, profoundly impacting patients' daily interactions and quality of life [1,4]. Approximately 95% of VS cases are unilateral and sporadic, presenting no clear etiological factors, whereas the remaining 5% are bilateral, stemming from neurofibromatosis type 2 (NF2) due to mutations in the NF2 gene [5,6]. The current therapeutic landscape for VS, aimed at enhancing patient quality of life and longevity, encompasses observation, minimally invasive surgery, and stereotactic radiotherapy, albeit without a definitive optimal approach [1,5].

Oxidative stress, which is marked by the overproduction of reactive oxygen species (ROS) compare to the strength of antioxidant defense, significantly contributes to oncogenesis. It modulates various cellular functions, including proliferation, apoptosis, and DNA integrity, potentially fostering tumor growth and therapeutic resistance. Hence, a deeper understanding of oxidative stress's role in cancer may open avenues for new targeted therapies that leverage cancer cells' susceptibility to ROS-induced damage [7,8]. However, the investigation into oxidative stress's specific impact on VSs remains scant.

Oxidative stress is increasingly acknowledged as a pivotal element in the pathogenesis, advancement, and therapeutic outcomes of a variety of cancer forms, including breast cancer, hepatocellular carcinoma in the liver, and lung cancer [9–11]. In the context of breast cancer, oxidative stress is intricately linked to the tumor's aggressiveness, metastatic capacity, and resistance to chemotherapeutic agents, underscoring the potential for therapeutic strategies aimed at mitigating oxidative stress. In hepatocellular carcinoma, the role of oxidative stress extends beyond promoting tumor cell proliferation and survival; it is also a key player in facilitating cell migration and invasion, thereby serving as both a marker for prognosis and a target for therapy. Likewise, in lung cancer, oxidative stress significantly influences tumor dynamics by modifying cell metabolism, growth, and apoptosis, which are integral to the efficacy of treatments and the overall prognosis of patients, particularly with regard to targeted therapies and immunotherapies. The complex involvement of

oxidative stress across these cancer types underscores its critical role in deciphering disease mechanisms and formulating novel therapeutic approaches, positioning it as a prime focus for future research endeavors aimed at enhancing cancer care and patient outcomes.

Extensive research on oxidative stress in malignant tumors has illuminated critical molecular mechanisms, such as the regulatory functions of ROS in cell proliferation, apoptosis, and DNA damage. These findings lay a groundwork for understanding the potential role of oxidative stress in VS, as VS may be influenced by similar oxidative stress pathways. Notably, even benign tumors like VS can experience growth promotion or regulation through oxidative stress, with moderate levels potentially activating signaling pathways that enhance cell proliferation. Given the demonstrated potential of antioxidant therapies in certain malignant cancers, there is a rationale for investigating similar antioxidant approaches in VS treatment research, aiming to uncover non-invasive therapeutic strategies for tumor management.

Notably, the exploration of oxidative stress in VS is markedly underdeveloped. This stands in stark contrast to the extensive body of research addressing oxidative stress mechanisms in various other cancer types, where significant advancements have been made. Presently, the realm of bioinformatics research experiences a pronounced void in studies focused on the oxidative stress responses within these tumors. Initiating research in this domain could unlock pivotal insights into the distinctive nature of VSs, laying the groundwork for the identification of innovative therapeutic avenues. Consequently, a thorough examination of the oxidative stress response in VSs is not only crucial for demystifying their pathogenesis but also promises to significantly influence the crafting of groundbreaking treatment methodologies.

The objective of this research is to elucidate the function of oxidative stress reactions in the pathology of VS by leveraging bioinformatics methodologies. Our investigation includes identifying differentially expressed genes (DEGs) and developing ceRNA and protein-protein interaction (PPI) networks to isolate hub genes. Furthermore, it incorporates comprehensive analytical methods including gene ontology (GO), kyoto encyclopedia of genes and genomes (KEGG), as well as gene set enrichment analysis (GSEA), coupled with differential expression analysis of critical genes, Gene Set Variation Analysis (GSVA), and analysis of immune infiltration. The research further extends to the construction of interaction networks for mRNA-RBP, mRNA-drug, and mRNA-TF. Central to this study is the discovery of a biomarker associated with the disease, providing groundbreaking insights for its diagnosis and treatment strategies.

## 2 Materials and methods

### 2.1 Data acquisition

Gene expression profiles of GSE54934 [12] and GSE56597 [13] were retrieved from the Gene Expression Omnibus (GEO) database using the R package GEOquery [14]. Both datasets originate from human nerve tissue; GSE54934 was based on platform GPL6244, and GSE56597 on GPL10739. Each dataset included 31 VS and 9 control samples. A summary of platform, tissue, species, and sample counts for each dataset is provided in Table 1.

**Table 1. GEO microarray chip information.**

|  | GSE54934 | GSE56597 |
|---|---|---|
| Platform | GPL6244 | GPL10739 |
| Type | Array | Array |
| Species | Homo sapiens | Homo sapiens |
| Tissue | Nerve | Nerve |
| Samples in Tumor group | 31 | 31 |
| Samples in Control group | 9 | 9 |
| Reference | PMID: 25333347 | PMID: 25533176 |

Oxidative stress-related genes (OSRGs) were obtained from the GeneCards [15] and MSigDB [16] databases. In GeneCards, 412 protein-coding OSRGs with a relevance score > 5 were selected. From MSigDB, several oxidative stress-related gene sets were extracted, including responses via VHL and oxidative stress-induced senescence pathways, yielding 358 genes. After combining and removing duplicates, 683 OSRGs were retained (S1 Table).

Batch effects between GSE54934 and GSE56597 were removed using the sva package [17], resulting in a merged dataset with 62 VS and 18 control samples. Gene expression matrices were normalized and annotated using the limma package [18]. Principal Component Analysis (PCA) analysis [19] was conducted before and after batch correction to assess normalization efficacy.

## 2.2 Identification of VS-associated oxidative stress-related differentially expressed genes (OSRDEGs)

Based on the merged GEO dataset, samples were categorized into VS and control groups. Differential expression analysis was performed using the limma package [18], with the criteria set at |logFC| > 1.5 and adjusted p-value < 0.05. Genes with logFC > 1.5 were considered upregulated, and those with logFC < −1.5 were considered downregulated. The Benjamini-Hochberg method was applied to control the false discovery rate.

To identify oxidative stress-related differentially expressed genes (OSRDEGs), DEGs from the combined dataset were intersected with the 683 previously identified OSRGs. The overlapping genes were defined as OSRDEGs. These genes were visualized using a Venn diagram and a heatmap generated by the pheatmap package.

## 2.3 GO and KEGG pathway enrichment analysis

Functional enrichment analysis of the OSRDEGs was performed using the clusterProfiler R package [20]. GO terms for Biological Process (BP), Cellular Component (CC), and Molecular Function (MF) were examined [21], and KEGG pathway enrichment was assessed [22].

Significant terms were selected based on an adjusted p-value < 0.05 and false discovery rate (FDR, q-value) < 0.25, with p-values corrected using the Benjamini-Hochberg method.

## 2.4 Gene set enrichment analysis (GSEA)

GSEA [23] was performed to evaluate whether predefined gene sets were differentially associated with VS phenotypes. Genes from the combined GEO dataset were ranked by logFC, and GSEA was conducted using the clusterProfiler R package.

GSEA parameters were set with an initialization seed of 2023 and 1000 permutations. The gene sets analyzed ranged from 10 to 500 genes, as defined in the MSigDB database [16]. Significant gene sets were selected based on an adjusted p-value < 0.05 and FDR (q-value) < 0.25.

## 2.5 PPI network and hub gene screening

A PPI network for OSRDEGs was constructed using the STRING database [24], with a moderate confidence interaction score threshold of 0.400. High-density regions within the network were selected for further analysis, as they likely represent molecular complexes.

The CytoHubba plugin for Cytoscape [25,26] was used to rank genes based on five algorithms: Maximal Clique Centrality (MCC), Degree, Maximum Neighborhood Component (MNC), Edge Percolated Component (EPC), and Closeness [27]. The top ten genes were selected for further investigation, and a Venn diagram was generated to identify common genes across algorithms. These genes were designated as hub genes related to oxidative stress.

## 2.6 Construction of regulatory network

The regulatory network for hub genes was constructed by retrieving transcription factors (TFs) from the ChIPBase database [28]. TF-gene interactions were filtered for those with more than six samples in both the upstream and downstream directions.

Additionally, the relationship between hub genes and miRNAs was analyzed using the StarBase v3.0 database [29]. miRNA-gene interactions were selected based on data from at least three sources, and the mRNA-miRNA regulatory network was visualized using Cytoscape.

## 2.7 The differential expression of hub genes

To assess the differential expression of hub genes between the VS and control groups, we conducted a comparative analysis using the pROC R package. Receiver Operating Characteristic (ROC) curves were generated to evaluate the diagnostic value of hub gene expression in distinguishing VS from control samples.

The Area Under the Curve (AUC) was calculated for each gene. AUC values range from 0.5 to 1, with values closer to 1 indicating superior diagnostic performance. AUC scores between 0.5 and 0.7 suggest low accuracy, between 0.7 and 0.9 moderate accuracy, and above 0.9 high accuracy.

## 2.8 Immune infiltration analysis (ssGSEA)

Immune cell infiltration was analyzed using the ssGSEA method [30], which quantifies the relative abundance of various immune cells. The relative abundance of immune cell types, such as activated CD8+T cells, dendritic cells, and natural killer cells, was calculated for each sample.

The ggplot2 R package was used to visualize the distribution of immune cells between VS and control groups. Significant differences in immune cell composition were identified, and a correlation analysis was performed using the Spearman rank correlation method. Heatmaps and bubble plots were generated using the pheatmap and ggplot2 packages to visualize the correlation between hub genes and immune cells.

## 2.9 Statistical analysis

Data analysis was performed using R software (version 4.3.0). For comparisons between two groups, the Student's t-test was used for normally distributed data, while the Mann-Whitney U test was used for non-normally distributed data.

For comparisons across three or more groups, the Kruskal-Wallis test was applied. Correlations between molecular entities were calculated using the Spearman rank correlation coefficient. All statistical tests were two-tailed, with a significance level set at $p < 0.05$.

## 2.10 Ethics statement

This study does not involve human participants, human data, or animals; therefore, institutional ethical approval and consent were not required.

## 3 Results

### 3.1 Technology roadmap

Fig 1 summarizes the analytical workflow for this study. Briefly, we merged two VS transcriptomic datasets (GSE54934 and GSE56597) and removed batch effects using the sva package; quality control was assessed by distribution boxplots and PCA before and after debatching. Differential expression was computed with limma (|logFC| > 1.5; adjusted $p < 0.05$), yielding 336 DEGs in the combined dataset. Intersecting these DEGs with curated oxidative-stress–related genes produced 15 OSRDEGs. We then performed GO/KEGG enrichment, constructed a PPI network to prioritize hub genes (CytoHubba), built TF–mRNA and miRNA–mRNA regulatory networks, validated hub-gene expression and diagnostic performance via ROC analysis, and profiled immune-cell infiltration using ssGSEA (Fig 1).

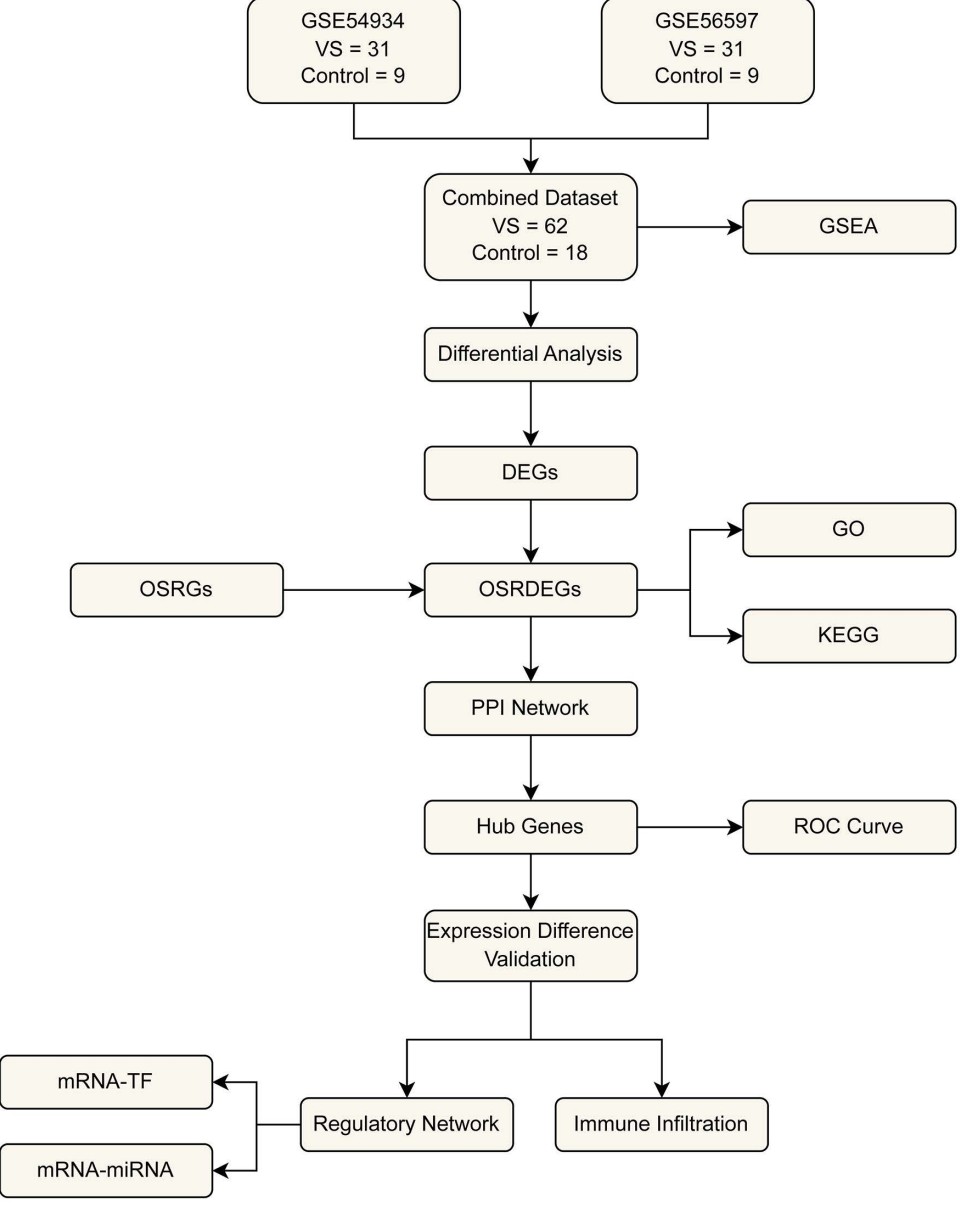

**Fig 1. Overview of the analytical workflow.** The flowchart summarizes the main steps of the study, including data preprocessing, differential expression analysis, functional enrichment, network construction, and validation analyses.

### 3.2 Merging of VS datasets

Firstly, the R package sva was employed to eliminate batch effects on GSE54934 and GSE56597, resulting in the creation of combined dataset(s). Subsequently, the distribution boxplot (Fig 2A-B) served to analyze and contrast the expression values present within the datasets, before and after the removal of batch effects. Additionally, PCA plot (Fig 2C-D) was used to compare the distribution of low-dimensional features both prior to and following the removal of batch effects. The distribution boxplots and PCA plots collectively illustrated the effective elimination of batch effects in the VS dataset samples following the removal process.

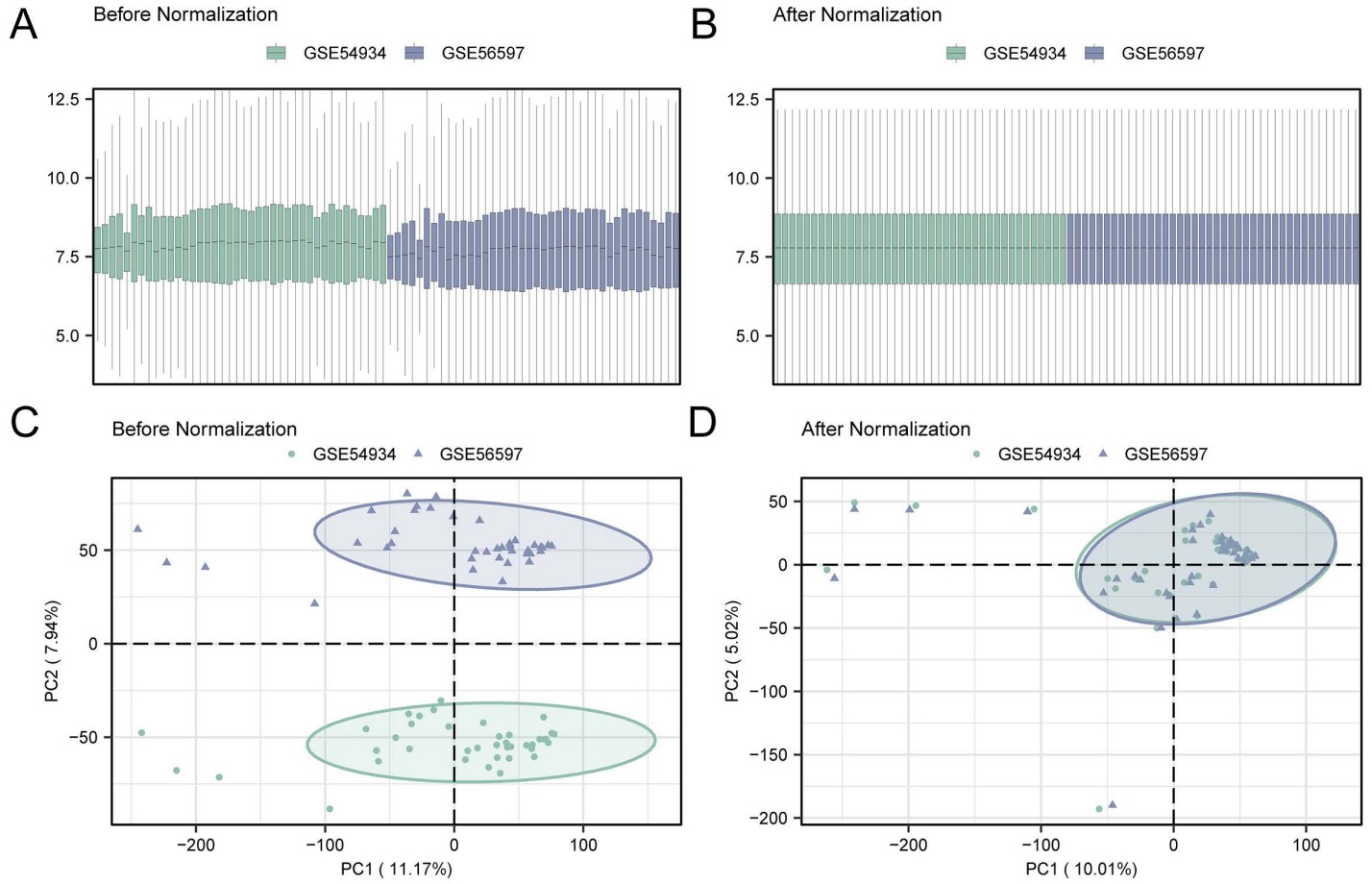

**Fig 2. Removal batch effects of GSE54934 and GSE56597.** A. Box plot illustrating the distribution of combined GEO datasets prior to batch effect removal. B. Post-batch integrated GEO datasets (combined datasets) distribution boxplots. C. 2D PCA plot of the datasets before debatching. D. 2D PCA plots of combined GEO datasets after debatching. Green is the VS dataset GSE54934, and blue is the VS dataset GSE56597.

### 3.3 VS-related oxidative stress–related DEGs

The data of combined GEO Datasets were segregated into VS group and control group. To investigate variations in gene expression levels between these groups, we performed the differential analysis using the R package limma. This analysis identified DEGs across the two datasets, yielding the following results: a total of 336 DEGs met the criteria of |logFC|>1.5 and adjusted p<0.05 in the combined datasets. Of these, 217 genes were upregulated (logFC>1.5 and adjusted p<0.05), and 119 genes were downregulated (logFC<−1.5 and adjusted p<0.05). The results of this variance analysis are illustrated in the volcano plot (Fig 3A).

In order to get the oxidative stress related differentially expressed genes (OSRDEGs), to be all | logFC |>1.5 and adjusted p<0.05 the DEGs, oxidative stress related genes (OSRGs) get intersection and map Venn(Fig 3B). Received 15 oxidative stress (OSRDEGs) differentially expressed genes, respectively is: *PLA2G2A, SELE, CAV1, EGFR, IL6, ADI-POQ, CPE, CYP1B1* is expressed, *AKR1C1, MGST1, CYBB, CDKN2A, CDH2, IL18, OLR1*. Based on the results from the intersection, we scrutinized the expression variations of OSRDEGs across distinct sample groups within combined dataset(s), and the results were visualized using a pheatmap created with the R package pheatmap (Fig 3C).

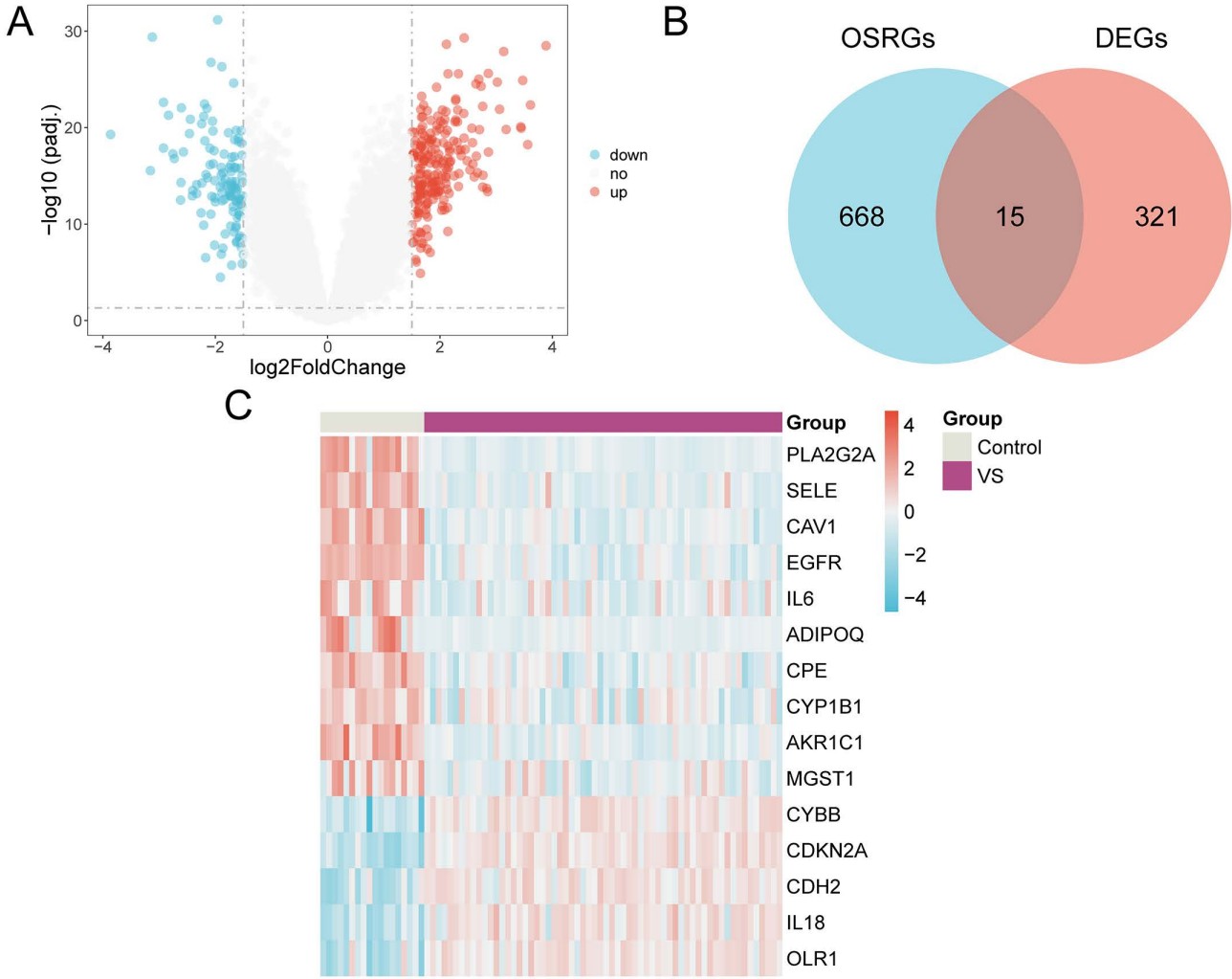

**Fig 3. Differential gene expression analysis.** A. Volcano plot of differentially expressed gene analysis of the VS group and the control group in combined dataset(s). B. DEGs in integrated GEO datasets (combined datasets), OSRGs venn diagram. C. Heat map of OSRDEGs in the integrated GEO datasets (combined datasets). The VS group is in purple, and the control group is in gray. In the heatmap, the color red corresponds to elevated expression levels, whereas blue corresponds to reduced expression levels, and the depth of color represents the degree of expression.

## 3.4 GO and pathway (KEGG) enrichment analysis

The BP, CC, MF and biological pathway (KEGG) of 15 OSRDEGs were further explored by GO and pathway (KEGG) enrichment analysis. The 15 OSRDEGs were employed for GO and pathway (KEGG) enrichment analysis, with specific results detailed in Table 2. The findings revealed that in VSs (VS), 15 differentially expressed genes related to oxidative stress (OSRDEGs) were primarily involved in leukocyte cell-cell adhesion. They also contribute to regulate various biological processes like peptidyl-tyrosine phosphorylation, cell-cell adhesion, macrophage derived foam cell differentiation, and are associated with CCs like secretory granules, rafts, and endocytic vesicles. Moreover, these genes exhibit diverse MFs including binding to sialic acid, carboxylic acids, organic acids, and cytokines, as well as interacting with ATPases. It was also enriched in lipid and atherosclerosis, african trypanosomiasis, malaria, cytochrome P450-mediated xenobiotic metabolism, chemical carcinogenesis-ROS and additional biological pathways

**Table 2. Result of GO and KEGG enrichment analysis for OSRDEGs.**

| ONTOLOGY | ID | Description | GeneRatio | BgRatio | p-value | p-adjust | q-value |
|---|---|---|---|---|---|---|---|
| BP | GO:0007159 | leukocyte cell-cell adhesion | 6/15 | 415/18614 | 5.00 e-07 | 6.10 e-04 | 2.42 e-04 |
| BP | GO:0050730 | regulation of peptidyl-tyrosine phosphorylation | 5/15 | 255/18614 | 1.25 e-06 | 6.10 e-04 | 2.42 e-04 |
| BP | GO:0022407 | regulation of cell-cell adhesion | 6/15 | 491/18614 | 1.34 e-06 | 6.10 e-04 | 2.42 e-04 |
| BP | GO:0010743 | regulation of macrophage derived foam cell differentiation | 3/15 | 32/18614 | 2.07 e-06 | 7.09 e-04 | 2.81 e-04 |
| BP | GO:0010742 | macrophage derived foam cell differentiation | 3/15 | 38/18614 | 3.51 e-06 | 7.44 e-04 | 2.95 e-04 |
| CC | GO:0030667 | secretory granule membrane | 5/15 | 319/19518 | 2.97 e-06 | 9.82 e-05 | 6.07 e-05 |
| CC | GO:0045121 | membrane raft | 5/15 | 323/19518 | 3.15 e-06 | 9.82 e-05 | 6.07 e-05 |
| CC | GO:0098857 | membrane microdomain | 5/15 | 324/19518 | 3.20 e-06 | 9.82 e-05 | 6.07 e-05 |
| CC | GO:0044853 | plasma membrane raft | 3/15 | 114/19518 | 8.39 e-05 | 1.93 e-03 | 1.19 e-03 |
| CC | GO:0030666 | endocytic vesicle membrane | 3/15 | 196/19518 | 4.15 e-04 | 7.64 e-03 | 4.72 e-03 |
| MF | GO:0033691 | sialic acid binding | 2/15 | 22/18369 | 1.42 e-04 | 1.77 e-02 | 8.25 e-03 |
| MF | GO:0031406 | carboxylic acid binding | 3/15 | 200/18369 | 5.25 e-04 | 2.57 e-02 | 1.20 e-02 |
| MF | GO:0043177 | organic acid binding | 3/15 | 212/18369 | 6.22 e-04 | 2.57 e-02 | 1.20 e-02 |
| MF | GO:0005125 | cytokine activity | 3/15 | 235/18369 | 8.39 e-04 | 2.60 e-02 | 1.22 e-02 |
| MF | GO:0051117 | ATPase binding | 2/15 | 86/18369 | 2.19 e-03 | 4.62 e-02 | 2.16 e-02 |
| KEGG | hsa05417 | Lipid and atherosclerosis | 5/15 | 215/8645 | 2.23 e-05 | 2.03 e-03 | 1.35 e-03 |
| KEGG | hsa05143 | African trypanosomiasis | 3/15 | 37/8645 | 3.17 e-05 | 2.03 e-03 | 1.35 e-03 |
| KEGG | hsa05144 | Malaria | 3/15 | 50/8645 | 7.89 e-05 | 3.37 e-03 | 2.24 e-03 |
| KEGG | hsa00980 | Metabolism of xenobiotics by cytochrome P450 | 3/15 | 78/8645 | 2.97 e-04 | 9.52 e-03 | 6.34 e-03 |
| KEGG | hsa05208 | Chemical carcinogenesis – reactive oxygen species | 4/15 | 223/8645 | 4.70 e-04 | 1.20 e-02 | 8.02 e-03 |

(KEGG). The findings from the GO and pathway (KEGG) enrichment analysis were illustrated using bar plots (Fig 4A) and bubble plots (Fig 4B).

Concurrently, a network diagram depicting the BP, CC, MF as well as biological pathway (KEGG) was constructed based on the GO and pathway (KEGG) enrichment analysis (Fig 4C-F). The lines connect the relevant molecules to the annotations of the entries, with bigger nodes indicating entries that contain more molecules.

### 3.5 GSEA

To assess how gene expression in integrated GEO datasets (combined datasets) influences VS conditions, GSEA was applied to scrutinize the full spectrum of gene expression and associated biological pathways within these combined datasets. GSEA revealed significant enrichment for interferon signaling, cilium-related programs, and immune activation pathways (Table 3; Fig 5A). Results are summarized in Table 3 and visualized in Fig 5. In the combined dataset, enrichment also appeared in Hippocampal Synaptogenesis and Neurogenesis (Fig 5B), HATs Acetylate Histones (Fig 5C), Regulation of TP53 Expression and Degradation (Fig 5D), Host–Pathogen Interaction of Human Coronaviruses—Autophagy (Fig 5E) and other biologically related functions and signaling pathways.

### 3.6 Construction of PPI network and screening of hub genes

Firstly, PPI analysis was performed, and a PPI network of 15 OSRDEGs was built based on the STRING database (Fig 6A). The results of PPI network showed that 14OSRDEGs were related, which were: *PLA2G2A, SELE, CAV1, EGFR, IL6, ADIPOQ, CYP1B1, AKR1C1, MGST1, CYBB, CDKN2A, CDH2, IL18* and *OLR1*. Subsequently, five different algorithms from the CytoHubba plug-in within the Cytoscape software were employed to perform calculations of 14 OSRDEGs scores, and the OSRDEGs were ranked in order according to the scores. The five algorithms used include maximum

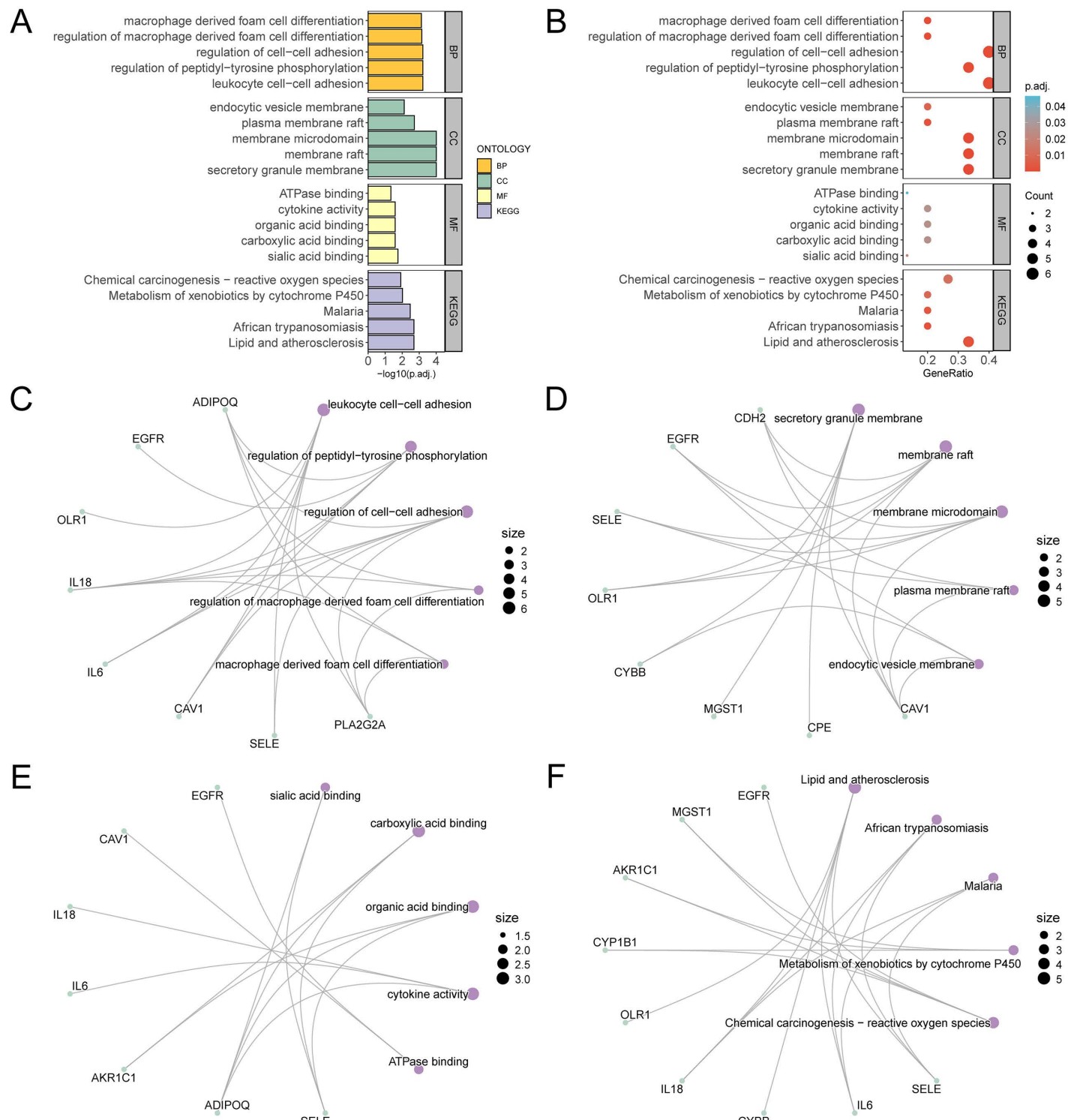

**Fig 4. GO and KEGG enrichment analysis for OSRDEGs.** A. OSRDEGs GO and pathway (KEGG) enrichment analysis results bar graph (A) and bubble plot (B) show: BP, CC, MF and biological pathway (KEGG), GO terms and KEGG terms are on the ordinate. C-F. GO and pathway (KEGG) enrichment analysis results of OSRDEGs: BP (C), CC (D), MF (E) and KEGG (F). Purple nodes denote items, green nodes denote molecules, with lines indicating the interactions between them. GO and KEGG pathway enrichment analysis were filtered with a stringent cutoff: an adjusted p of less than 0.05 and a q value signifying a false discovery rate threshold of 0.25. The p-value correction was performed using the Benjamini-Hochberg method.

**Table 3. Results of GSEA for combined datasets.**

| ID | setSize | EnrichmentScore | NES | p-value | p-adjust | q-value |
|---|---|---|---|---|---|---|
| REACTOME_INTERFERON_ALPHA_BETA_SIGNALING | 57 | 0.67184 | 2.51489 | 1.68 e-03 | 3.53 e-02 | 2.76 e-02 |
| REACTOME_INTERFERON_SIGNALING | 164 | 0.56046 | 2.50744 | 1.42 e-03 | 3.53 e-02 | 2.76 e-02 |
| WP_TYROBP_CAUSAL_NETWORK_IN_MICROGLIA | 56 | 0.66946 | 2.50413 | 1.64 e-03 | 3.53 e-02 | 2.76 e-02 |
| WP_GENES_RELATED_TO_PRIMARY_CILIUM_DEVELOPMENT_BASED_ON_CRISPR | 80 | 0.63072 | 2.46063 | 1.64 e-03 | 3.53 e-02 | 2.76 e-02 |
| WP_TYPE_I_INTERFERON_INDUCTION_AND_SIGNALING_DURING_SARSCOV2_INFECTION | 29 | 0.69294 | 2.24268 | 1.74 e-03 | 3.53 e-02 | 2.76 e-02 |
| WP_CILIARY_LANDSCAPE | 188 | 0.49208 | 2.22860 | 1.45 e-03 | 3.53 e-02 | 2.76 e-02 |
| KEGG_ALLOGRAFT_REJECTION | 25 | 0.69730 | 2.16582 | 1.78 e-03 | 3.53 e-02 | 2.76 e-02 |
| KEGG_LYSOSOME | 113 | 0.51461 | 2.16289 | 1.49 e-03 | 3.53 e-02 | 2.76 e-02 |
| REACTOME_INTERACTION_BETWEEN_L1_AND_ANKYRINS | 30 | 0.66562 | 2.15977 | 1.75 e-03 | 3.53 e-02 | 2.76 e-02 |
| WP_HOSTPATHOGEN_INTERACTION_OF_HUMAN_CORONAVIRUSES_INTERFERON_INDUCTION | 32 | 0.65376 | 2.15491 | 1.75 e-03 | 3.53 e-02 | 2.76 e-02 |
| KEGG_ASTHMA | 24 | 0.69892 | 2.14936 | 1.77 e-03 | 3.53 e-02 | 2.76 e-02 |
| REACTOME_ANTIVIRAL_MECHANISM_BY_IFN_STIMULATED_GENES | 72 | 0.55484 | 2.14932 | 1.62 e-03 | 3.53 e-02 | 2.76 e-02 |
| WP_IMMUNE_RESPONSE_TO_TUBERCULOSIS | 21 | 0.72106 | 2.14557 | 1.78 e-03 | 3.53 e-02 | 2.76 e-02 |
| WP_JOUBERT_SYNDROME | 68 | 0.55747 | 2.12862 | 1.63 e-03 | 3.53 e-02 | 2.76 e-02 |
| KEGG_RNA_DEGRADATION | 52 | 0.57496 | 2.11396 | 1.69 e-03 | 3.53 e-02 | 2.76 e-02 |
| WP_BARDETBIEDL_SYNDROME | 76 | 0.53165 | 2.07612 | 1.61 e-03 | 3.53 e-02 | 2.76 e-02 |
| WP_HIPPOCAMPAL_SYNAPTOGENESIS_AND_NEUROGENESIS | 26 | 0.58425 | 1.83514 | 3.52 e-03 | 3.65 e-02 | 2.85 e-02 |
| REACTOME_HATS_ACETYLATE_HISTONES | 89 | 0.45637 | 1.82785 | 1.56 e-03 | 3.53 e-02 | 2.76 e-02 |
| REACTOME_REGULATION_OF_TP53_EXPRESSION_AND_DEGRADATION | 32 | 0.52119 | 1.71793 | 5.26 e-03 | 4.58 e-02 | 3.59 e-02 |
| WP_HOSTPATHOGEN_INTERACTION_OF_HUMAN_CORONAVIRUSES_AUTOPHAGY | 18 | 0.59620 | 1.68458 | 3.55 e-03 | 3.65 e-02 | 2.85 e-02 |

neighborhood component (MNC), maximal clique centrality (MCC), degree, edge percolated component (EPC), closeness. Then, TOP 10 OSRDEGs from the five algorithms were used to draw PPI networks, namely MCC (Fig 6B), Degree (Fig 6C), MNC (Fig 6D), EPC (Fig 6E) and Closeness (Fig 6F). Among them, the color of the circles from red to yellow represents the score from high to low. Finally, the common genes identified by all five algorithms were selected and the Venn diagram (Fig 6G) was created for analytical purpose. The intersection gene from the algorithm was identified as the hub genes of VS, and the 9 hub genes were: *IL6, CYBB, CAV1, EGFR, SELE, IL18, CDKN2A, ADIPOQ, CDH2.*

### 3.7 Construction of regulatory network

Firstly, hub genes were obtained combined with TFs through ChIPBase database, and the screening conditions were number of sample found (upstream + downstream) > 6. The mRNA-TF regulatory network was created and displayed utilizing cytoscape software (Fig 7A). Among them, there were 7 hub genes and 26 TFS, with specific details provided in S2 Table.

Then, the miRNA related to the hub genes were obtained through the StarBase database, and the screening criteria were mRNA-miRNA interaction relationships recorded from at least three sources. The mRNA-miRNA Regulatory network was developed and depicted utilizing Cytoscape software (Fig 7B). Among them, there are 6 hub genes and 56miRNAs, with specific details outlined in S3 Table.

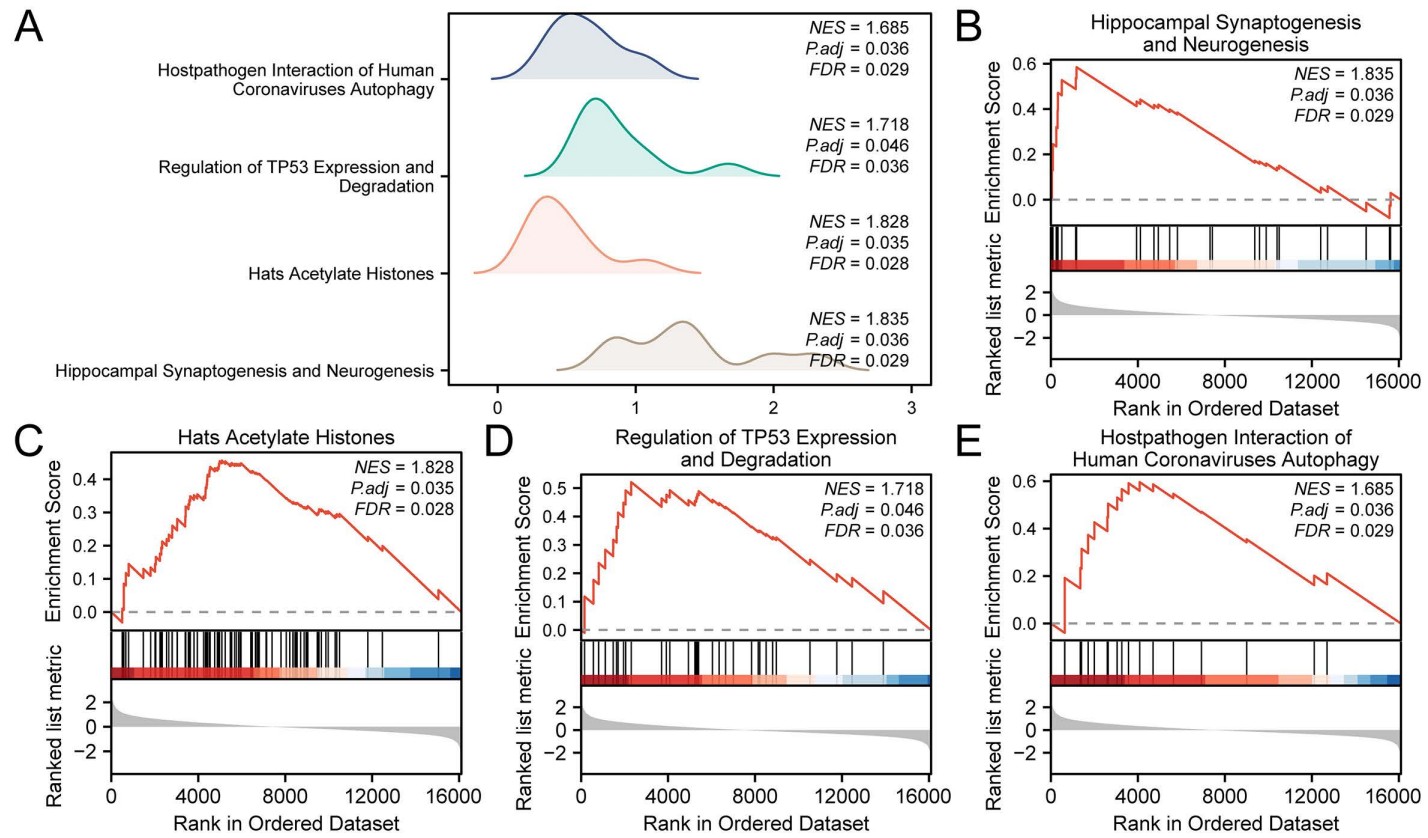

**Fig 5. GSEA for combined datasets.** A. Ridge plots of representative gene sets. B–E. Enriched programs: Hippocampal Synaptogenesis and Neurogenesis (B); HATs Acetylate Histones (C); Regulation of TP53 Expression and Degradation (D); Host–Pathogen Interaction of Human Coronaviruses—Autophagy (E).

### 3.8 Differential expression verification and ROC curve analysis of hub genes

To explore the expression differences of hub genes in combined GEO datasets, the group comparison figure (Fig 8A) shows the difference analysis results of the expression profiles of 9 hub genes in the VS group and the control group in the integrated GEO datasets (combined datasets). The differential results displayed (Fig 8A) that the expression profiles of 9 hub genes in both the VS group and the control group from the combined datasets demonstrated highly significant statistical variances ($p < 0.001$). These genes are: *IL6, CYBB, CAV1, EGFR, SELE, IL18, CDKN2A, ADIPOQ* and *CDH2*. Finally, ROC curves were generated using the R package pROC, utilizing the expression profiles of hub genes in combined dataset(s). The ROC curve (Fig 8B-J) illustrated that the expression patterns of hub genes (*CYBB, CAV1, EGFR, SELE, IL18, CDKN2A* and *CDH2*) demonstrated high precision (AUC > 0.9) in differentiating the VS group from the control group; the expression levels of IL6 and ADIPOQ exhibited a degree of precision (0.7 < AUC < 0.9) in the classification of VS group and control group.

### 3.9 Analysis of immune infiltration in VSs

The expression matrices of the combined datasets were applied to compute the immune cell infiltration levels for 28 types of immune cells by the ssGSEA algorithm. Firstly, group comparison plots illustrated the variations in the expression of immune cell infiltration abundances across different groups. Group comparison diagram (Fig 9A) showed that all 17

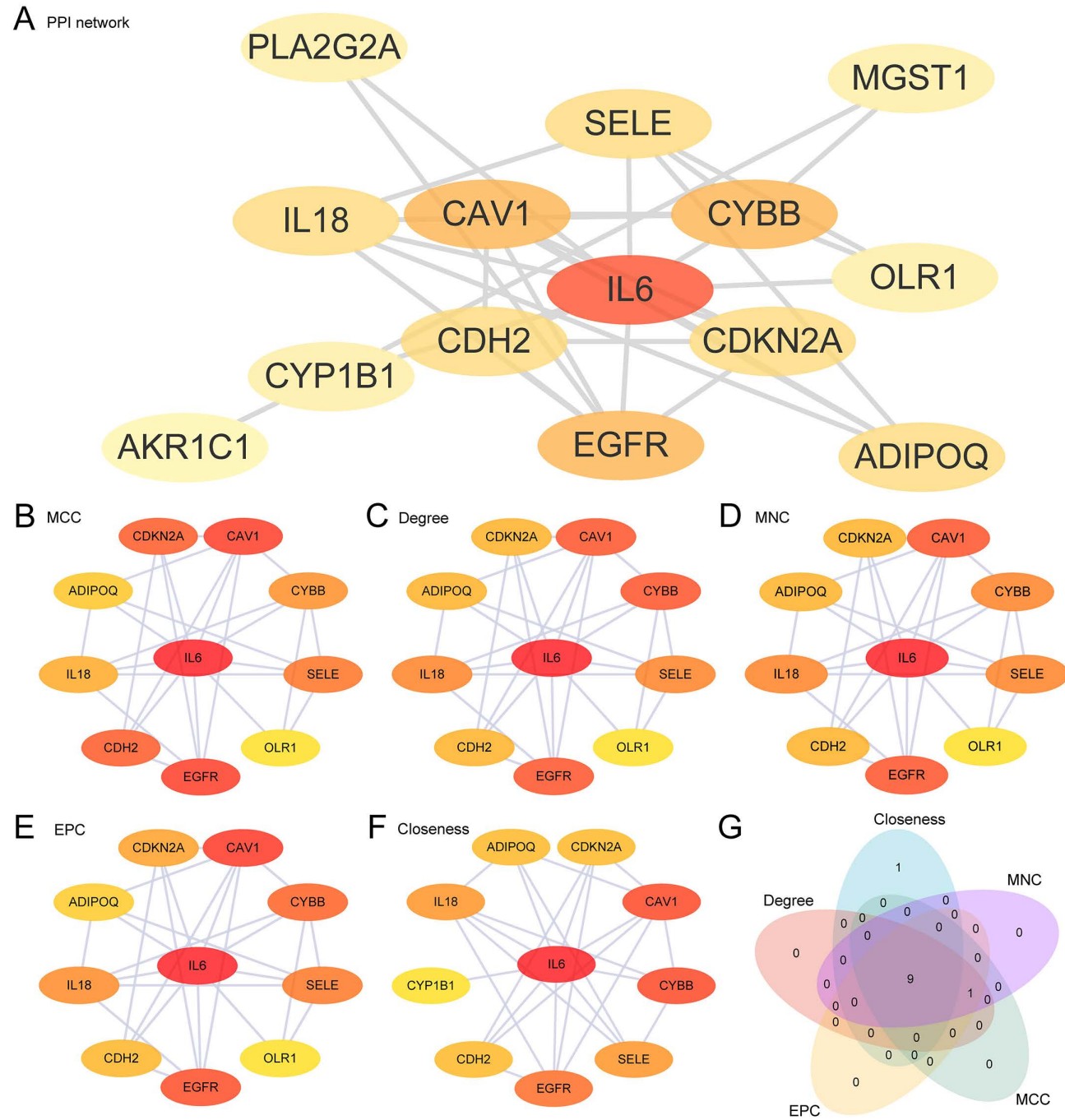

**Fig 6. Analysis of PPI network and hub genes.** A. PPI network of OSRDEGs derived from STRING database. B-G. PPI network of TOP 10 OSR-DEGs calculated using five algorithms from cytohubba plug-in, including MCC (B), degree (C), MNC (D), EPC (E) and closeness (F). G. OSRDEGs Venn diagram of TOP 10 for the 5 algorithms of the Cyto Hubba plugin.

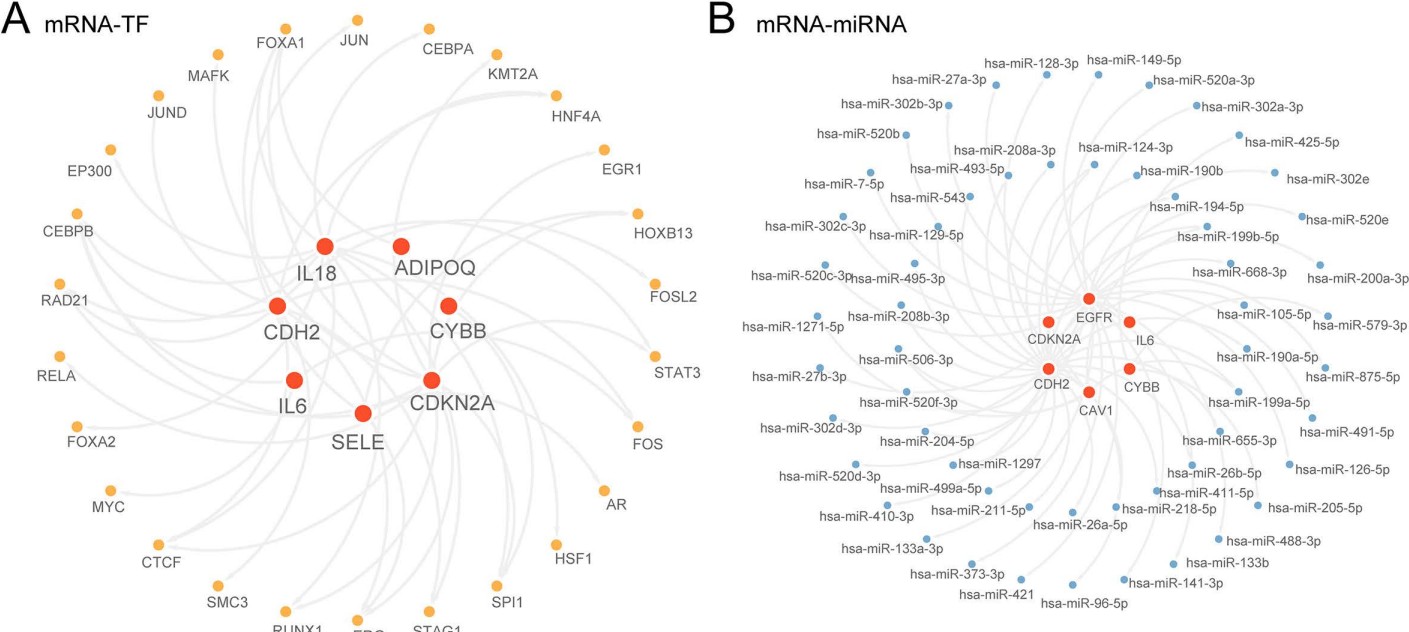

**Fig 7. Regulatory network analysis of hub genes.** A-B. mRNA-TF regulatory network (A) and mRNA-miRNA regulatory network (B) of hub genes. TF, transcription factors. Red shows hub genes, orange shows TFs, and blue shows miRNAs.

immune cells showed statistical significance (p value < 0.05), namely: Activated CD4 + T cell, Activated CD8 + T cell, Activated dendritic cell, CD56bright natural killer cell, CD56dim natural killer cell, Effector memory CD4 + T cell, Effector memory CD8 + T cell, Immature B cell, Immature dendritic cell, Mast cell, Natural killer cell, Neutrophil, Plasmacytoid dendritic cell, Regulatory T cell, T follicular helper cell, Type 1 T helper cell, Type 17 T helper cell. Then, the correlation heatmap displayed the correlation findings for the infiltration abundance of 17 immune cells in combined dataset(s) (Fig 9B). The findings indicated that the majority of immune cells exhibited strong positive correlations, with the Type 1 T helper cell and Mast cell demonstrating the most significant positive correlation(r value = 0.825, p value < 0.05). Finally, the relationship linking hub genes to immune cell infiltration levels was shown by correlation bubble plot (Fig 9C). The results of correlation bubble plot showed that most of the immune cells showed strong positive correlation, and the gene CYBB and immune cell Regulatory T cell had the strongest significant positive correlation (r value = 0.888, $P$ value < 0.05).

## 4 Discussion

VS is a benign tumors derived from Schwann cells, characterized by slow growth and low invasiveness. They typically present with Antoni A and B areas, with Verocay bodies as pathognomonic features. In contrast, breast cancer, lung cancer, and hepatocellular carcinoma are malignant tumors with higher growth rates and invasiveness [9–11]. Unlike VS, these cancers can metastasize to other body parts. VS rarely metastasize and have a low malignant potential, with features such as improved cellularity and mitotic figures being indicative of malignancy [31].

The research into VS is of paramount importance due to their significant incidence rate. Although classified as benign, the profound impact on patient well-being, including hearing loss, balance difficulties, and in critical cases, life-threatening complications, cannot be overstated. VS, often located in the cerebellopontine angle, can cause severe symptoms and have a poor prognosis due to their location and growth. These tumors can compress the facial nerve (cranial nerve VII) and the trigeminal nerve (cranial nerve V), leading to facial weakness, numbness, and pain. Larger tumors may press on

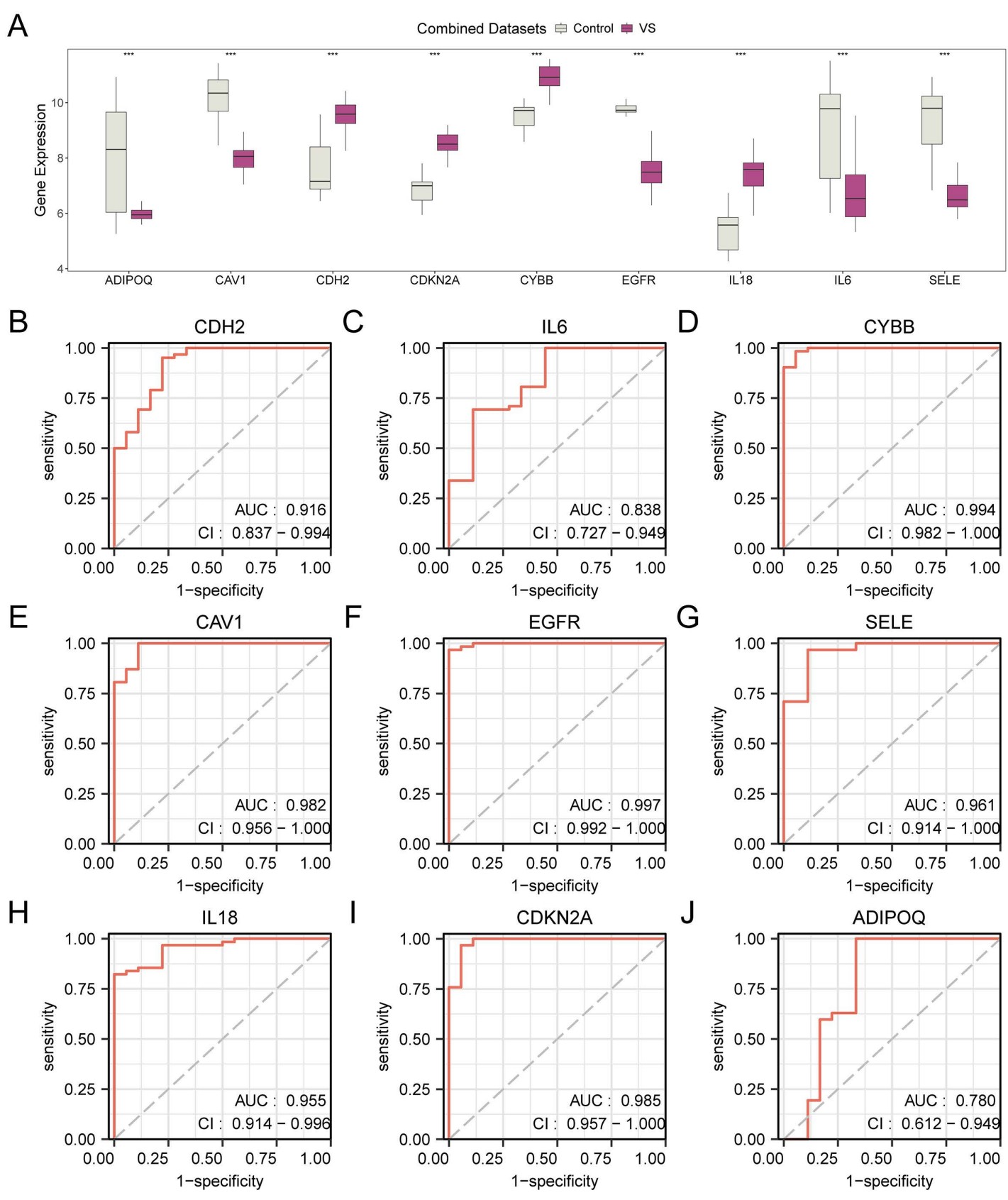

**Fig 8. Differential expression validation and ROC curve analysis.** A. Group comparison plot of hub genes in VS group and control group of combined GEO datasets. B–J. ROC curves of hub genes in the integrated GEO datasets (combined datasets). *** represents a p < 0.001 and highly statistically significant. AUC values were interpreted as follows: 0.5–0.7 low, 0.7–0.9 moderate, and >0.9 high diagnostic accuracy. TPR, true positive rate; FPR, false positive rate. The color coding in the diagrams: gray is used to denote the control group, while purple signifies the VS group.

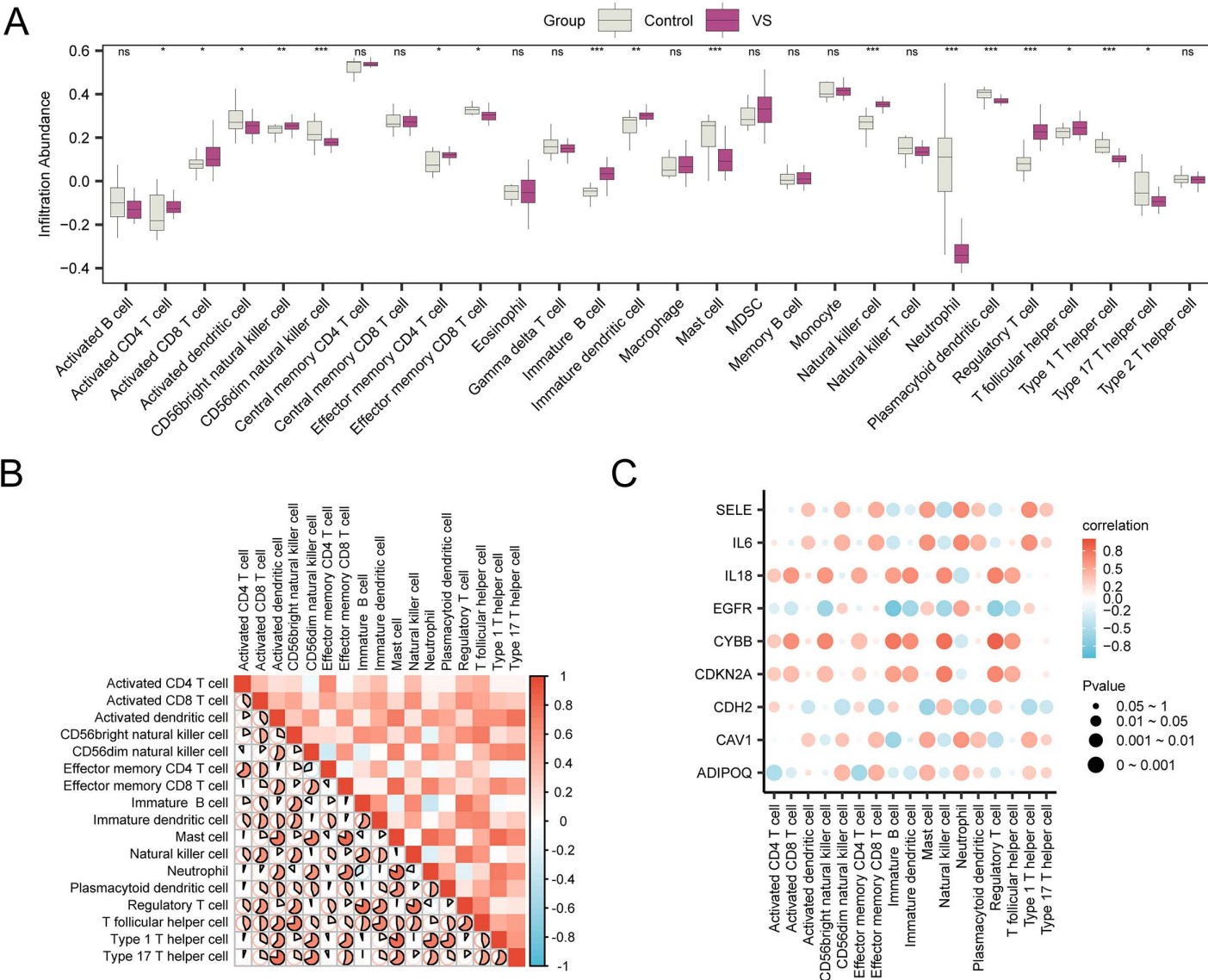

**Fig 9. Immune infiltration analysis by ssGSEA algorithm.** A. Group comparison plot of immune cells in VS group and control group of combined GEO datasets. B. Correlation heatmap of immune cell infiltration abundance in the integrated GEO datasets (combined datasets). C. Bubble chart depicting the relationship correlating hub genes and immune cell infiltration levels in the integrated GEO Datasets. (combined datasets). ssGSEA, single-sample gene-set enrichment analysis; VS, vestibular schwannoma. ns stands for p value ≥ 0.05, not statistically significant; * represents p value < 0.05, indicating statistical significance; ** represents p value < 0.01, indicating a high level of statistical significance; *** denotes p value < 0.001, signifying an even higher degree of statistical significance. An absolute value of correlation coefficient (r value) less than 0.3 indicate either weak or no association, 0.3 to 0.5 suggest weak correlation, 0.5 to 0.8 denote a moderate correlation, and exceeding 0.8 signifies a robust correlation. Control group (gray), VS group (purple). A positive association is signified by the color red, whereas blue denotes a negative association. The color's saturation corresponds to the strength of the correlation.

the brainstem, causing ataxia, hydrocephalus, and potentially life-threatening brainstem compression [1,4]. The proximity to critical neural structures like the facial nerve and brainstem can lead to significant morbidity. Surgical and radiosurgical treatments aim to minimize damage to these structures while controlling tumor growth. The potential for these complications call for an enriched comprehension of its underlying pathogenesis. Additionally, the constraints posed by current treatment modalities, predominantly invasive surgeries accompanied by significant morbidity, accentuate the pressing need for innovative therapeutic avenues. Thus, this investigation stands at the forefront of advancing our understanding of VS, especially in revealing novel biomolecular pathways and exploring viable non-surgical treatment alternatives, thereby enhancing patient outcomes and shedding light on the biological mechanisms of akin neurotumors. Given the complex nature and the considerable impact of VS on patient quality of life, further research into this condition is imperative to uncover new insights and develop more effective, less invasive treatment strategies.

Oxidative stress, marked by a discord between ROS formation and the body's detoxification capability, is a key factor in the pathology of diverse ailments, including neurodegenerative diseases, cardiovascular disorders, and cancers [9–11,32,33]. Within the realm of VS, oxidative stress emerges as a critical element contributing to tumor proliferation and progression. Echoing previous findings, our study elucidates the impact of oxidative stress on vital cellular undertakings such as DNA damage, apoptosis, and inflammation, which are central to tumor biology. A deeper understanding of oxidative stress's specific impact on VS may herald the advent of targeted antioxidant therapies, offering patients therapeutic alternatives that are both less invasive and more effective. This research significantly bridges an existing gap in literature, concentrating on the influence of oxidative stress on tumor development, thereby furnishing fresh insights into the cellular and molecular mechanisms propelling VS. These discoveries not only signify a substantial academic advancement in disease comprehension but also bear direct practical implications. By delineating pathways modulated by oxidative stress, this research establishes a foundation for the creation of focused therapies, potentially revolutionizing the clinical management of VS, considering the disease's complexity and variability.

In the pursuit of a comprehensive bioinformatics landscape, this study has identified 15 OSRDEGs in VS, including prominent genes such as *EGFR, IL6, CAV1*, and *CDKN2A*. These genes are implicated in crucial biological processes and pathways, illuminating the complex pathogenesis of VS through the perspective of oxidative stress. In various malignancies, these genes have been linked to tumorigenesis, progression, and therapeutic responsiveness [34–36], indicating potential similarities in molecular mechanisms between VS and other cancers, thereby paving pathways for therapeutic intervention. Notably, several of the hub genes we identified—CDKN2A, CYBB, and ADIPOQ—have not been previously reported in relation to VS. Their discovery in this context suggests novel molecular associations between oxidative stress and VS pathogenesis. For instance, CDKN2A, a cell cycle regulator known for its role in high-grade gliomas and melanoma, has not been implicated in VS, marking it as a potentially novel biomarker. Similarly, CYBB, commonly linked to oxidative burst in phagocytes, and ADIPOQ, associated with metabolic and inflammatory signaling, have not been connected to schwannoma biology. These findings highlight unexplored pathways and may provide starting points for future mechanistic or therapeutic investigations in VS. In our study, nine hub genes were identified, among which IL6, EGFR, CAV1, and CDKN2A are notable due to their established involvement in various tumors. IL6 has been previously implicated in VS, where it contributes to inflammatory signaling and correlates with tumor progression and cyst formation [37]. EGFR overexpression has been documented in schwannoma tissue and is associated with proliferative activity, suggesting its potential as a therapeutic target [38]. CAV1 has also been reported in relation to Schwann cell signaling and myelination, and its dysregulation may reflect a role in schwannoma pathophysiology [39]. However, CDKN2A, while widely studied in malignancies like melanoma and glioblastoma, has not yet been reported in the context of VS to our knowledge. We therefore highlight CDKN2A as a novel and potentially important finding in the context of VS pathogenesis.

The GO and KEGG enrichment analyses have unveiled valuable insights into the roles of OSRDEGs, associating them with immune-related processes and cellular adhesion. These insights suggest a possible connection between immune dysregulation and VS pathogenesis. Moreover, the emphasis on pathways concerning lipid metabolism and ROS

accentuates the significance of metabolic alterations and oxidative stress in the tumor biology of VS, offering novel perspectives for therapeutic targeting.

GSEA has disclosed significant enrichment in pathways linked to hippocampal synaptogenesis and neurogenesis, histone acetylation, and the regulation of TP53 expression, among others. These pathways hint at the potential influence of oxidative stress on neural development and epigenetic mechanisms within VS, possibly impacting tumor initiation and progression. Additionally, the enrichment in pathways associated with host-pathogen interactions and autophagy underlines the intricate interplay between viral infections, immune response, and VS, suggesting new therapeutic and diagnostic opportunities.

The establishment of PPI networks and the identification of hub genes such as IL6, EGFR, and CDKN2A underscore their central role in the pathophysiology of VS. These genes, recognized for their significance across various cancers, emphasize the probable influence of oxidative stress and inflammatory responses in the development of VS. The analysis of transcription factor (TF) and microRNA (miRNA) regulatory networks has unveiled a complex network of regulatory interactions, pinpointing the intricate control mechanisms at play in the pathogenesis and progression of VS.

The confirmation of key gene expression differences and ROC curve analysis has substantiated the diagnostic potential of several hub genes, offering insights into novel biomarkers and therapeutic targets. The significant expression differences and the high accuracy in ROC analysis reinforce their role in the molecular mechanisms of VS and their feasibility as diagnostic markers.

The immune infiltration analysis conducted via ssGSEA has revealed a distinct immune landscape within the VS microenvironment, characterized by differential infiltration of diverse immune cell classes. This research highlights the pivotal function of the immune milieu in driving the development and spread of VS, illustrating the complex interplay of immune cells and identifying prospective goals for immunological intervention. Our ssGSEA-based immune infiltration analysis revealed marked differences in several immune cell types between VS and control tissues, particularly in regulatory T cells (Tregs), activated CD4＋/CD8＋T cells, and natural killer cells. These findings align partially with previous histological studies showing the presence of immune infiltrates in VS, particularly CD3+ and CD8＋T lymphocytes [40], as well as increased cytokine activity in cystic VS [37]. However, the detailed immune landscape and correlation with oxidative stress signatures have not been systematically characterized before. Our data bridge this gap by uncovering potential cross-talk between OSRGs such as CYBB and IL6, and immune activation patterns, suggesting a role for oxidative stress in shaping the immunological microenvironment of VS. This cross-disciplinary insight may open avenues for immunomodulatory or anti-inflammatory therapeutic strategies in treating VS.

It is crucial to recognize that this investigation pioneers the application of bioinformatics to analyze the impact of oxidative stress on the development of VS. Prior to this, research into the molecular underpinnings of this condition had not explicitly focused on the oxidative stress pathways. By utilizing a comprehensive bioinformatics approach to identify and analyze DEGs associated with oxidative stress, our study breaks new ground in the molecular exploration of VS. This pioneering effort not only opens up new avenues for understanding the disease but also sets a foundational platform for subsequent studies to investigate these complex biological interactions further.

Despite the strengths of this study, several limitations should be acknowledged. First, our findings are based solely on publicly available transcriptomic data, which may lack sufficient clinical annotation and sample diversity. Second, while we identified key OSRGs and pathways using robust computational methods, no experimental validation (e.g., qPCR, western blotting, or immunohistochemistry) was performed to confirm their functional roles in VS. Third, the immune infiltration analysis inferred from bulk RNA-seq data may not fully capture the complexity of the tumor microenvironment. Future studies should incorporate larger, multi-center cohorts and in vitro or in vivo experiments to validate the biological relevance and therapeutic potential of the hub genes identified. Overall, this study provides a novel systems-level perspective on the pathogenesis of VS by integrating oxidative stress-related gene expression, immune cell infiltration, and multi-level regulatory networks. To our knowledge, this is the first bioinformatics investigation to

systematically link oxidative stress signaling to the immunological microenvironment of VS, uncovering genes such as CDKN2A, CYBB, and CAV1 as potential mediators. Furthermore, hub genes like EGFR and IL6, already implicated in other cancers, may also represent druggable targets in VS that warrant further validation. These findings offer new hypotheses for the development of non-surgical, mechanism-based therapies, particularly in cases where current treatment options are limited or contraindicated.We recommend expanding the sample size and conducting experimental validations in future studies to further explore the roles of these genes in VS. To build upon these findings, future research should focus on experimental validation of the hub genes identified, particularly CDKN2A, EGFR, IL6, and CYBB, using cell-based assays, gene knockdown/overexpression models, and immunohistochemistry on patient tissues. Investigating the role of these genes in modulating immune cell infiltration, especially regulatory T cells and cytotoxic T lymphocytes, may elucidate novel immunopathological mechanisms in VS. In addition, studies should assess the predictive and prognostic value of these genes in larger, prospectively collected clinical cohorts. Exploring the feasibility of targeted inhibitors or anti-inflammatory agents (e.g., anti-IL6 antibodies or EGFR inhibitors) could open new therapeutic avenues. Integrating multi-omics data (proteomics, single-cell RNA-seq) and fostering interdisciplinary collaboration with clinical and translational teams will be essential to transform these molecular insights into practical interventions.

Furthermore, we believe that interdisciplinary collaboration and close ties with clinical researchers will be crucial in advancing this field. By integrating basic research, clinical trials, and drug development, we are confident that we can more effectively address the therapeutic challenges of VS. We look forward to future studies building on these findings to develop more effective treatment methods.

## 5 Conclusion

In summary, our integrative bioinformatics analysis suggests that oxidative stress may contribute to vestibular schwannoma progression through its involvement in apoptosis, ROS regulation, and immune modulation. We identified nine hub genes, including CDKN2A, CYBB, and EGFR, which may serve as potential biomarkers or therapeutic candidates. While these findings provide a useful resource for hypothesis generation, they require future experimental validation to establish their functional significance and clinical applicability.

## Supporting information

**S1 Table. Oxidative stress–related genes (OSRGs) used for downstream intersection.** This table lists 683 unique human gene symbols curated for oxidative-stress relevance and used to intersect with DEGs in this study. Columns: gene_symbol (HGNC-approved symbol; one per row). File format: CSV, UTF-8, comma-separated; 683 rows × 1 column. Notes: Species = Homo sapiens; duplicates removed; see Methods for curation criteria.
(CSV)

**S2 Table. Predicted mRNA–TF regulatory pairs included in network analyses.** This table contains 46 nonredundant edges linking candidate mRNAs to putative transcription factors used to build the TF–mRNA network. Columns: mRNA (HGNC symbol of target gene), TF (HGNC symbol of transcription factor). File format: CSV, UTF-8; 46 rows × 2 columns. Notes: One row per directed pair (TF→mRNA); only pairs retained after filtering described in Methods.
(CSV)

**S3 Table. Predicted mRNA–miRNA regulatory pairs included in network analyses.** This table contains 60 nonredundant edges linking candidate mRNAs to putative microRNAs used to build the miRNA–mRNA network. Columns: mRNA (HGNC symbol of target gene), miRNA (miRNA ID). File format: CSV, UTF-8; 60 rows × 2 columns. Notes: One row per directed pair (miRNA→mRNA); only pairs retained after filtering described in Methods.
(CSV)

## Author contributions

**Conceptualization:** Yubin Xue.

**Data curation:** Yubin Xue, Mingyue Wang.

**Formal analysis:** Yubin Xue.

**Methodology:** Yubin Xue, Mingyue Wang.

**Supervision:** Yubin Xue.

**Validation:** Hongwei Ma.

**Visualization:** Mingyue Wang.

**Writing – original draft:** Yubin Xue.

**Writing – review & editing:** Yubin Xue, Hongwei Ma.

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
