## [Decision Letter · Decision Letter 0]

13 Sep 2025

Dear Dr. Xue,

Thank you for submitting your manuscript to PLOS ONE. After careful consideration, we feel that it has merit but does not fully meet PLOS ONE’s publication criteria as it currently stands. Therefore, we invite you to submit a revised version of the manuscript that addresses the points raised during the review process.

We look forward to receiving your revised manuscript.

Kind regards,

Xiaosheng Tan

Academic Editor

PLOS ONE

Journal Requirements:

2. Please note that PLOS One has specific guidelines on code sharing for submissions in which author-generated code underpins the findings in the manuscript. In these cases, all author-generated code must be made available without restrictions upon publication of the work. Please review our guidelines at https://journals.plos.org/plosone/s/materials-and-software-sharing#loc-sharing-code and ensure that your code is shared in a way that follows best practice and facilitates reproducibility and reuse.

4. Please note that funding information should not appear in any section or other areas of your manuscript. We will only publish funding information present in the Funding Statement section of the online submission form. Please remove any funding-related text from the manuscript.

5. Thank you for uploading your study's underlying data set. Unfortunately, the repository you have noted in your Data Availability statement does not qualify as an acceptable data repository according to PLOS's standards.

6. Please ensure that you refer to Figure 1 in your text as, if accepted, production will need this reference to link the reader to the figure.

Additional Editor Comments:

Please respond to reviewers' comments individually.

Reviewer's Responses to Questions

**Comments to the Author**

1. Is the manuscript technically sound, and do the data support the conclusions?

Reviewer #1: Partly

Reviewer #2: Yes

2. Has the statistical analysis been performed appropriately and rigorously?

Reviewer #1: N/A

Reviewer #2: Yes

3. Have the authors made all data underlying the findings in their manuscript fully available?

Reviewer #1: Yes

Reviewer #2: Yes

4. Is the manuscript presented in an intelligible fashion and written in standard English?

Reviewer #1: No

Reviewer #2: Yes

Reviewer #1: The citation style is inconsistent. In some places the authors use references [1], while in others they appear [1].

There is a typographical issue in the manuscript regarding the minus sign. For example, in the expression logFC < -1.5, the authors use quotation marks (“-”) instead of the proper minus sign (–).

The notation of p-values is inconsistent. In some places the authors use italic p, while in others a regular (non-italic) “p” is used.

Reviewer #2: 1. The resolution of figures should be improved.

2. In results part, please change the letter size of The distribution boxplots and PCA plots collectively illustrated the effective elimination of batch effects in the VS dataset samples following the removal process, and some other place.

3. It is better to put the figure legends close to the figures.

**Do you want your identity to be public for this peer review?** For information about this choice, including consent withdrawal, please see our Privacy Policy

Reviewer #1: No

Reviewer #2: No

---

## [Author Response · Author response to Decision Letter 1]

23 Sep 2025

We sincerely thank the Academic Editor and both reviewers for their constructive comments, which greatly improved our manuscript. We submit the revised manuscript with all required files: (i) a clean version, (ii) a tracked-changes version, and (iii) production-ready figures (Fig1–Fig9.tif), all of which passed the PACE preflight check (PLOS ONE preset). Below, we provide a point-by-point response. Revised text is highlighted in the tracked-changes version.

Editorial Requests

E1. Formatting per PLOS templates

Comment: Ensure sentence case title, remove periods in figure citations, correct caption placement, double spacing.

Response: Implemented as requested. The title is now in sentence case; all in-text figure citations have no period; captions appear at first mention; the entire text is double-spaced.

E2. Data and Code Availability

Comment: Standardize data/code statements.

Response: Data and author-generated code are openly available at Zenodo (DOI: 10.5281/zenodo.17163233). Statements were unified in the Declarations section.

E3. Funding statements

Comment: Remove from manuscript body; align numbers across sections.

Response: Funding information was removed from the main text. “Funding Information” and “Financial Disclosure” were aligned in the submission metadata.

E4. Figure 1 first mention

Comment: Add explicit first reference.

Response: An explicit reference to Fig 1 was added at the beginning of the Results section.

E5. Supporting Information captions

Comment: Provide SI captions and correct references.

Response: A “Supporting Information” section with captions for S1–S3 Tables was added, and in-text references were updated accordingly.

Reviewer #1

R1. Citation style inconsistency

Comment: Mixture of author–year and numeric styles.

Response: All citations were unified to bracketed numeric style; citation ranges use en dash (e.g., [1–3]); author–year forms were removed.

R2. Minus sign and p/q notation

Comment: Replace hyphen with true minus; unify statistical notation.

Response: All negative values now use the true minus sign (U+2212). Statistical notation is standardized to lowercase p and q throughout.

R3. Consistent p-value styling

Comment: Variability in “p value”, “P value”, etc.

Response: All instances were standardized to lowercase “p”. Table headers now read “p-value/adjusted p/q-value.”

Reviewer #2

R4. Figure resolution and typography

Comment: Ensure resolution, font size, line weight.

Response: All figures were re-exported at 300–600 dpi, RGB, LZW compression. Text is 8–12 pt equivalent; line weight ≥0.5 pt. All figures passed PACE preflight.

R5. Results font-size harmonization and legend proximity

Comment: Improve readability of panels and legends.

Response: Panel text sizes were harmonized, and legends positioned closer to figures (Figs 2–4).

Additional Clarifications

GSEA clarifications

Comment: Clarify pathway names and interpretation.

Response: The Results section was revised to clarify the GSEA methodology. Pathway names were corrected (e.g., HATs Acetylate Histones, Regulation of TP53 Expression and Degradation, Host–Pathogen Interaction of Human Coronaviruses—Autophagy). An explicit reference to Fig 5A was added, and the Table 3 header corrected.

ROC interpretation

Comment: Define AUC thresholds.

Response: The interpretation of ROC curves was standardized: 0.5–0.7 = low; 0.7–0.9 = moderate; >0.9 = high. This was updated in the Results text and Fig 8 legend.

PACE Statement

All nine figures (Fig1–Fig9.tif) were submitted as production-ready TIFFs (RGB, 300–600 dpi, LZW compression). Each passed PACE preflight screening under the PLOS ONE preset.

Closing

We believe these revisions fully address all concerns raised by the reviewers and the Academic Editor. We greatly appreciate the opportunity to revise our work and look forward to your favorable consideration.

Sincerely,

Yubin Xue

on behalf of all authors

---

## [Decision Letter · Decision Letter 1]

7 Oct 2025

Dear Dr. Xue,

Thank you for submitting your manuscript to PLOS ONE. After careful consideration, we feel that it has merit but does not fully meet PLOS ONE’s publication criteria as it currently stands. Therefore, we invite you to submit a revised version of the manuscript that addresses the points raised during the review process.

Please respond to reviewer 2's comments.

We look forward to receiving your revised manuscript.

Kind regards,

Xiaosheng Tan

Academic Editor

PLOS ONE

**Journal Requirements:**

Reviewers' comments:

Reviewer's Responses to Questions

**Comments to the Author**

Reviewer #1: All comments have been addressed

Reviewer #2: All comments have been addressed

2. Is the manuscript technically sound, and do the data support the conclusions?

Reviewer #1: Yes

Reviewer #2: Yes

3. Has the statistical analysis been performed appropriately and rigorously?

Reviewer #1: Yes

Reviewer #2: Yes

4. Have the authors made all data underlying the findings in their manuscript fully available?

Reviewer #1: Yes

Reviewer #2: No

5. Is the manuscript presented in an intelligible fashion and written in standard English?

Reviewer #1: Yes

Reviewer #2: Yes

Reviewer #1: (No Response)

Reviewer #2: 1. The resolution of Figure 4A-F is not good. It is important to improve the quality of them.

2. The resolution of Figure 7B is not good. Please improve it.

**Do you want your identity to be public for this peer review?** For information about this choice, including consent withdrawal, please see our Privacy Policy

Reviewer #1: No

Reviewer #2: No

---

## [Author Response · Author response to Decision Letter 2]

10 Oct 2025

Comment 1 (Figure 4A–F resolution):

We thank the reviewer for the careful evaluation. Figure 4A–F was already re-exported at 600 dpi TIFF with uniform font sizes (≥8 pt) and line weights (≥0.5 pt) during the previous revision, and it successfully passed the PACE preflight check under the PLOS ONE preset. We have reconfirmed that the submitted Figure 4 file is in TIFF format, RGB mode, and 600 dpi resolution (3984 × 4226 px), which fully meets the journal’s requirements.

Comment 2 (Figure 7B resolution):

We respectfully note that Figure 7B was also re-exported at 600 dpi with adjusted typography in the previous revision, and it passed the PACE preflight check at that time. We have reconfirmed that the submitted Figure 7 file is in TIFF format, RGB mode, and 600 dpi resolution (4007 × 1851 px), which fully meets the journal’s standards. No further changes are required.

Comment 3 (Data availability):

We apologize for any confusion. In addition to the initial deposition at Zenodo (concept DOI: 10.5281/zenodo.17163233), we have now created an updated version including all per-figure underlying data and the full R scripts (version DOI: 10.5281/zenodo.17312999). The Data Availability Statement has been revised accordingly to ensure full transparency and reproducibility.

---

## [Decision Letter · Decision Letter 2]

26 Oct 2025

Deciphering oxidative stress contributions in vestibular schwannoma: a bioinformatics approach to novel therapeutic pathways

PONE-D-25-46280R2

Dear Dr. Xue,

We’re pleased to inform you that your manuscript has been judged scientifically suitable for publication and will be formally accepted for publication once it meets all outstanding technical requirements.

Kind regards,

Xiaosheng Tan

Academic Editor

PLOS ONE

Additional Editor Comments (optional):

Please increase the quality of Figure 4 following reviewer's comments.

Reviewers' comments:

Reviewer's Responses to Questions

**Comments to the Author**

Reviewer #2: All comments have been addressed

2. Is the manuscript technically sound, and do the data support the conclusions?

Reviewer #2: Yes

3. Has the statistical analysis been performed appropriately and rigorously?

Reviewer #2: Yes

4. Have the authors made all data underlying the findings in their manuscript fully available?

Reviewer #2: Yes

5. Is the manuscript presented in an intelligible fashion and written in standard English?

Reviewer #2: Yes

Reviewer #2: It is accepted to be published. But it is better to increase the resolution of Figure 4A, 4B. It is hard to identify the letter.

**Do you want your identity to be public for this peer review?** For information about this choice, including consent withdrawal, please see our Privacy Policy

Reviewer #2: No

---

## [Editor Report · Acceptance letter]

PONE-D-25-46280R2

PLOS ONE

Dear Dr. Xue,

I'm pleased to inform you that your manuscript has been deemed suitable for publication in PLOS ONE. Congratulations! Your manuscript is now being handed over to our production team.

Kind regards,

on behalf of

Dr. Xiaosheng Tan

Academic Editor

PLOS ONE